# Molecular detection of *Orientia tsutsugamushi* infection in bats from the China-Myanmar border

Yun Long[1☉], Jiawei Tian[1☉], Peiyu Han[1☉], Song Wu[1], Lidong Zong[1], Chenjie He[1], Yuhong Chen[1], Wanchun Cao[1], Bo Wang[2], Lijun Guo[1]*, Yunzhi Zhang [1]*

**1** Yunnan Key Laboratory of Screening and Research on Anti-Pathogenic Plant Resources from Western Yunnan, Key Laboratory for Cross-Border Control and Quarantine of Zoonoses in Universities of Yunnan Province, Institute of Preventive Medicine, School of Public Health, Dali University, Dali, Yunnan, China, **2** Programme in Emerging Infectious Diseases, Duke-NUS Medical School, Outram, Central Region, Singapore

☉ These authors contributed equally to this work.
* glj72588@163.com (LG); zhangyunzhi1818@163.com (YZ)

## Abstract

*Orientia tsutsugamushi* (Ot), the causative agent of tsutsugamushi disease (TD), has been detected in *Muridae*, *Soricidae*, *Cricetidae*, *Canidae*, *Artiodactyla*, and birds, as well as in its trombiculid mite vectors. However, to date, scarce reports have documented Ot infection in bats. TD is an ancient zoonotic disease transmitted through the bite of infected trombiculid mites. The global disease burden of TD, particularly in impoverished regions, warrants renewed attention and a reevaluation of public health strategies. In this study, we analyzed bat samples for Ot using nested PCR (nPCR) and real-time quantitative PCR (qPCR) for both qualitative and quantitative detection. Genetic evolution and divergence time of the detected Ot sequences were assessed using bioinformatics tools, including BioAider, Clustal X2, MEGA-X, and BEAST. Ot was detected in 7.32% (44/601) of bat samples by qPCR. At least three genotypes, including Karp, Gilliam, and Kato, were identified in both insectivorous bats (*Hipposideros larvatus* and *Hipposideros armiger*) and frugivorous bats (*Rousettus amplexicaudatus* and *Cynopterus sphinx*). Ot DNA was detected in multiple tissues, including heart, kidney, spleen, lung, rectum, liver, and brain, with median copy numbers ranging from 28.60 to 1069.76 copies/µL. Notably, divergence analysis suggests that Ot isolated from bats emerged around 126 AD, later than its appearance in rodents, humans, and chiggers (approximately 4140 BC), indicating that Ot infection in bats may originate from other animals or vectors. Our findings recommend ongoing monitoring of Ot in bats and their ectoparasites, which will provide a basis for risk assessment and guide strategies for the prevention and control of scrub typhus.

**Data availability statement:** All data generated and analysed in this study are included in the manuscript and its Supporting information files. All sequences referenced in this article are retrievable from the NCBI database (https://www.ncbi.nlm.nih.gov, accessed on May 10, 2025). All sequences generated in this study have been submitted to GenBank (accession numbers: 56kDa TSA: PV719510–PV719519; 47kDa htrA: PV719504–PV719509; GroEL: PV719520–PV719523; 16S rRNA: PV697021–PV697023; Cytb: PV719524–PV719533). These data are publicly available through the accession numbers above.

**Funding:** This work was supported by the National Natural Science Foundation of China (No. U2002218 to YZ); Yunnan Health Training Project of High Level Talents (No. L-2017027 to YZ); Cross-border Control and Quarantine Innovation Group of Zoonosis of Dali University (No. ZKPY2019302 to YZ). The funders had no role in study design, data collection and analysis, decision to publish, or preparation of the manuscript.

**Competing interests:** The authors have declared that no competing interests exist.

## Author summary

*Orientia tsutsugamushi* (Ot) is an obligate, intracellular bacterium and the causative agent of tsutsugamushi disease (TD) in humans. Ot is vectored by the biting of the larval life stage of infected mites. Scarce previous studies have reported Ot infection in bats. In this study, we identified at least three genotypes of Ot (Karp, Gilliam, and Kato) in four bat species, including *H. larvatus*, *H. armiger*, *R. amplexicaudatus*, and *C. sphinx*, along the China-Myanmar border. Molecular clock analysis suggests that Ot strains isolated from bats emerged long after the divergence of strains in rodents, humans, and chiggers, supporting the hypothesis that Ot isolated from bats likely originated from established mammalian or vector reservoirs. Our findings underscore the need for ongoing surveillance of Ot in bats and their associated ectoparasites to inform risk assessments and guide scrub typhus prevention and control strategies.

## Introduction

Tsutsugamushi disease (TD) is a zoonotic illness caused by *Orientia tsutsugamushi* (Ot) that is primarily transmitted through the bite of infected trombiculid mite larvae [1,2], with rare non-vector routes such as transplacental transmission, transfusion, and organ transplantation [3–7]. Ot is a Gram-negative, obligate intracellular bacterium belonging to the class *Alphaproteobacteria*, order *Rickettsiales*, family *Rickettsiaceae*, and genus *Orientia*.

In nature, Ot is maintained and transmitted by trombiculid mites, which act as both reservoirs and vectors [1,8]. These mites undergo seven life stages: egg, deutovum (prelarva), larva, nymphochrysalis, nymph, imagochrysalis, and adult. Transmission to mammals, including humans, occurs via the bite of infected larvae [9]. Human-to-human transmission of scrub typhus, via routes such as transplacental transmission [7], transfusion [6], and organ transplantation [4], has previously been reported. Following exposure, patients typically develop symptoms within 5–14 days, including nonspecific flu-like symptoms, fever, headache, and eschar at the bite site [1,10]. In severe cases, TD can lead to multi-organ failure [1].

Several genes have been used as PCR targets for Ot detection, including the 56-kDa type-specific antigen gene (56-kDa TSA), 47-kDa high-temperature requirement A gene (47-kDa *htrA*), 60-kDa heat shock protein gene (GroEL), and 16S ribosomal RNA gene (16S rRNA) [11–13]. Among these, the 56-kDa TSA gene has become the most widely used for genotyping, given its high variability [14]. The 47-kDa *htrA* gene is more conserved and useful for broader detection [15], while the GroEL gene helps differentiate *Orientia* from *Rickettsia* species [12]. The 16S rRNA gene remains the gold standard for taxonomic classification of prokaryotes [16]. Common Ot genotypes include Karp, Kato, Gilliam, TA763, TA686, Shimokoshi, Kawasaki, and Saitama [17].

Scrub typhus is one of the most underdiagnosed and underreported diseases globally, often requiring hospitalization, as recognized by the World Health

Organization (WHO) (https://www.who.int/publications/i/item/who-recommended-surveillance-standards, accessed on 20 May 2025). It is endemic to the so-called "tsutsugamushi triangle", which spans northern Japan and far eastern Russia in the north, to northern Australia in the south, and westward to Pakistan and Afghanistan [18,19]. However, the global distribution of scrub typhus appears to be wider than previously known, as *Orientia* species have been detected in the Middle East, Africa, and Chile [20–23]. Notably, *Orientia chuto* has been detected in southwestern Asia and eastern Africa [20,24,25].

In southern China, the high-risk areas for TD include Yunnan, Guangxi, Guangdong, Hainan, and Fujian Provinces. A 2019 study estimated that approximately 163 million people in these regions live in areas at potential risk of infection [26]. In Yunnan Province, both the number of cases and the incidence rate have continued to rise in recent years [27]. In neighboring Myanmar, serological evidence shows that scrub typhus is widespread, particularly in the central and northern areas [28], and is a major cause of acute febrile illness along the Myanmar border [29,30].

Research has demonstrated a broad range of potential hosts for Ot [31,32]. The systematic review by Elliott et al. [32] underscores the critical role of small mammals. These mammals maintain chigger populations, which are the primary Ot reservoir, and thereby become exposed to and infected with Ot when bitten. The review also describes various methods, including serology and PCR, for detecting Ot in vertebrates. Specifically, rodents from the family *Muridae*, as the primary dead-end hosts for Ot, show a high infection prevalence of 25.5%. Beyond rodents, other mammalian groups also demonstrated notable infection rates, including *Soricidae* (shrews) at 13.2%, and *Canidae* (primarily dogs) at 17.8%. Among the *Artiodactyla*, cows, goats, and pigs were tested only by serological methods, with an overall positive rate of 3.6%. Studies have indicated that birds can be infected with Ot [31–33], and can be parasitized by vector chiggers [34–36]. Elliott et al. mentioned in their study that *Chiroptera* were tested only by serological methods, and 12% of *Eptesicus serotinus* and 11% *Rhinolophus ferrumequinum* were positive [32]. However, a study in the United States investigating *Orientia* antibodies in *Eptesicus fuscus* found no evidence of the pathogen [37].

Bats (order *Chiroptera*) represent the second most diverse group of mammals after rodents [38]. As flying mammals capable of long-distance seasonal migration, bats are increasingly coming to contact with humans and domestic animals, especially as urbanization accelerates and they adapt to urban environments [39,40]. Despite their ecological importance and species richness, scarce studies have reported Ot infection in bats. In this study, we report the detection of multiple Ot genotypes in bats along the China-Myanmar border.

## Results

### Sample collection and detection

A total of 601 bat samples representing 4 families and 11 species were collected from Ruili City, Yingjiang County, and Gengma County along the China-Myanmar border. The dominant species included Schreiber's long-fingered bat (*Miniopterus schreibersi*, n = 208), Great Himalayan leaf-nosed bat (*Hipposideros armiger*, n = 119), Leschenault's fruit bat (*Rousettus leschenaultii*, n = 86), and Greater short-nosed fruit bat (*Cynopterus sphinx*, n = 73) (Table 1). Using qPCR, Ot DNA was detected in 44 samples, yielding an overall prevalence of 7.32% (95% CI: 5.23%–9.41%). The highest prevalence was observed in the Intermediate leaf-nosed bat (*Hipposideros larvatus*), with 30.00% positivity (n = 6, 95% CI: 8.00%–52.00%), followed by *Cynopterus sphinx* at 13.70% (n = 10, 95% CI: 5.62%–21.78%). The nPCR detected Ot in 10 samples, corresponding to an overall prevalence of 1.66% (95% CI: 0.64%–2.69%). The detection rate of qPCR was significantly higher than that of nPCR ($\chi^2 = 22.41$, $P < 0.001$).

### Sequence identity analysis of 56-kDa TSA, 47-kDa *htrA*, GroEL, and 16S rRNA genes

We successfully amplified and sequenced multiple genes of Ot from the qPCR-positive bat samples. A total of 10 Ot strains were amplified for the 56-kDa TSA gene, including four full-length sequences (1,602–1,620 bp) derived from NBB39, NBB41, NBB92, and RLB163 (GenBank accession numbers: PV719510, PV719511, PV719515, and PV719519),

**Table 1. Detection of *Orientia tsutsugamushi* from bats in this study.**

| Family | Species | Locations | Composition (%) | Prevalence (%) | | | | |
|---|---|---|---|---|---|---|---|---|
| | | | | qPCR | nPCR | | | |
| | | | | | 56-kDa TSA | 47-kDa *htrA* | GroEL | 16S rRNA |
| *Hipposideridae* | *Hipposideros armiger* | Yingjiang, Gengma | 119/601 (19.80) | 8/119 (6.72) | 3/119 (2.52) | 2/119 (1.68) | 1/119 (0.84) | 2/119 (1.68) |
| | *Hipposideros larvatus* | Yingjiang | 20/601 (3.33) | 6/20 (30.00) | 5/20 (25.00) | 4/20 (20.00) | 2/20 (10.00) | 1/20 (5.00) |
| *Pteropodidae* | *Cynopterus sphinx* | Yingjiang, Ruili | 73/601 (12.15) | 10/73 (13.70) | 1/73 (1.37) | 0/73 (0.00) | 0/73 (0.00) | 0/73 (0.00) |
| | *Macroglossus sobrinus* | Yingjiang | 1/601 (0.17) | 0/1 (0.00) | 0/1 (0.00) | 0/1 (0.00) | 0/1 (0.00) | 0/1 (0.00) |
| | *Rousettus leschenaultii* | Ruili | 86/601 (14.31) | 3/86 (3.49) | 0/86 (0.00) | 0/86 (0.00) | 0/86 (0.00) | 0/86 (0.00) |
| | *Rousettus amplexicaudatus* | Ruili | 42/601 (6.99) | 1/42 (2.38) | 1/42 (2.38) | 0/42 (0.00) | 1/42 (2.38) | 0/42 (0.00) |
| | *Eonycteris spelaea* | Ruili | 19/601 (3.16) | 0/19 (0.00) | 0/19 (0.00) | 0/19 (0.00) | 0/19 (0.00) | 0/19 (0.00) |
| | *Megaerops niphanae* | Ruili | 3/601 (0.50) | 0/3 (0.00) | 0/3 (0.00) | 0/3 (0.00) | 0/3 (0.00) | 0/3 (0.00) |
| *Vespertilionidae* | *Miniopterus schreibersi* | Gengma | 208/601 (34.61) | 15/208 (7.21) | 0/208 (0.00) | 0/208 (0.00) | 0/208 (0.00) | 0/208 (0.00) |
| | *Miniopterus fuliginosus* | Gengma | 28/601 (4.66) | 1/28 (3.57) | 0/28 (0.00) | 0/28 (0.00) | 0/28 (0.00) | 0/28 (0.00) |
| *Rhinolophidae* | *Rhinolophus blythi* | Gengma | 2/601 (0.33) | 0/2 (0.00) | 0/2 (0.00) | 0/2 (0.00) | 0/2 (0.00) | 0/2 (0.00) |
| Total | | | 601/601 (100.00) | 44/601 (7.32) | 10/601 (1.66) | 6/601 (1.00) | 4/601 (0.67) | 3/601 (0.50) |

and six partial fragments (171–894 bp). Six strains yielded sequences of the 47-kDa *htrA* gene, with five full-length sequences (1,401 bp) and one partial fragment (768 bp). For the GroEL gene, four strains were sequenced, including one full-length sequence (1,668 bp) from NBB41 (GenBank accession number: PV719520) and three partial fragments (738–1,392 bp). Additionally, three full-length 16S rRNA gene sequences (1,459 bp) were obtained (S1 Table). The final lengths of all obtained sequences resulted from the assembly of multiple amplified fragments by DNA walking.

Sequence identity analysis revealed that the 56-kDa TSA gene sequences shared 79.49%–100.00% nucleotide (nt) identity and 68.75%–100.00% amino acid (aa) identity with known Ot strains in GenBank. Specifically, strains NBB39, NBB41, NBB42, NBB45, and NBB52 from *H. larvatus* in Yingjiang County exhibited the highest nt identity (99.05%–100.00%) to the human isolate OTSg62 (KF777346) from India. Similarly, strains NBB92 and NBB127 from *H. armiger* in Yingjiang County were closely related (97.08%–98.13% nt identity) to the patient isolate XYP3 (ON745806) from Yunnan Province, China, while NBB104 from *H. armiger* in Yingjiang County aligned best (98.25% nt identity) with isolate XYP7 (ON745808), also from Yunnan Province, China. Notably, RLB107 from *C. sphinx* in Ruili City shared 97.95% nt identity with XYP4 (ON745807), and RLB163 from *R. amplexicaudatus* in Ruili City had 94.69% nt identity with the Indian patient sample 0809aTw (MW495817) (S1 Fig).

The 47-kDa *htrA* gene sequences displayed high similarity, with 95.84%–99.36% nt identity and 95.74%–100.00% aa identity compared reference Ot strains. Strains from *H. larvatus* in Yingjiang County (NBB39, NBB41, NBB42, NBB52) showed the closest affinity (99.22%–99.36% nt identity) to the Thai isolate CRF27 (HM156047), while NBB92 and NBB127 from *H. armiger* in Yingjiang County matched 98.72% nt identity with Indian isolate JJOtsu7 (CP166957) (S2 Fig).

The GroEL gene sequences exhibited 95.26%–99.57% nt identity and 98.39%–100.00% aa identity relative to published Ot sequences. NBB41, NBB42 from *H. larvatus* in Yingjiang County, and RLB163 from *R. amplexicaudatus* in Ruili City showed the highest nt identity (97.60%–97.83%) with the Japanese strain Kuroki (JX188394). Meanwhile, NBB92 from *H. armiger* in Yingjiang County had the highest nt identity (99.57%) with isolate SH205(KC688332) from a Korean field mouse (*Apodemus peninsulae*) (S3 Fig).

For the 16S rRNA gene, the three full-length sequences shared 98.22%–99.79% nt identity with GenBank references. NBB41 from *H. larvatus* and NBB92 and NBB127 from *H. armiger* in Yingjiang County aligned most closely (99.66%–99.79%) with the Ot Karp strain (NR_025860) from Japan (S4 Fig).

Collectively, our findings demonstrate a broad genetic diversity of Ot in bats from the China-Myanmar border region, with multiple strains exhibiting close genetic affinity to human clinical isolates from China, India, Thailand, Japan, and South Korea.

## Phylogenetic analysis

Phylogenetic trees based on the 56-kDa TSA gene revealed that NBB92, NBB104, and NBB127 from *H. armiger*, RLB163 from *R. amplexicaudatus*, and RLB107 from *C. sphinx* clustered with the Karp strain (AY956315) originally isolated in Taiwan, China. In contrast, NBB39, NBB41, NBB42, NBB45, and NBB52 from *H. larvatus* formed a distinct cluster with the Gilliam strain (DQ485289), also a human case in Taiwan, China (Fig 1A).

For the 47-kDa *htrA* gene, strains NBB39, NBB41, NBB42, and NBB52 from *H. larvatus*, along with NBB92 and NBB127 from *H. armiger*, grouped within the same clade as the Kato strain (HM595493), isolated from a Japanese patient (Fig 1B).

Phylogenetic analysis of the GroEL gene showed that NBB41 and NBB42 from *H. larvatus* and RLB163 from *R. amplexicaudatus* were closely related to the Kuroki strain (JX188394) from a Japanese patient. NBB92 from *H. armiger* clustered more closely with the Kato strain (JX188393), also from Japan. All GroEL sequences clustered within the Ot lineage from Japan and South Korea, clearly separated from the *Rickettsia* species (Fig 1C).

The phylogenetic tree based on the 16S rRNA gene showed that NBB41 from *H. larvatus* was mostly closely related to the Karp strain (NR_025860) from a Japanese patient. Meanwhile, NBB92 and NBB127 from *H. armiger* clustered with the Boryong strain (AM494475) isolated in South Korea. All sequences isolated from bats grouped with known Ot strains from Asia and Europe, and were distinctly separated from *Rickettsia* species (Fig 1D).

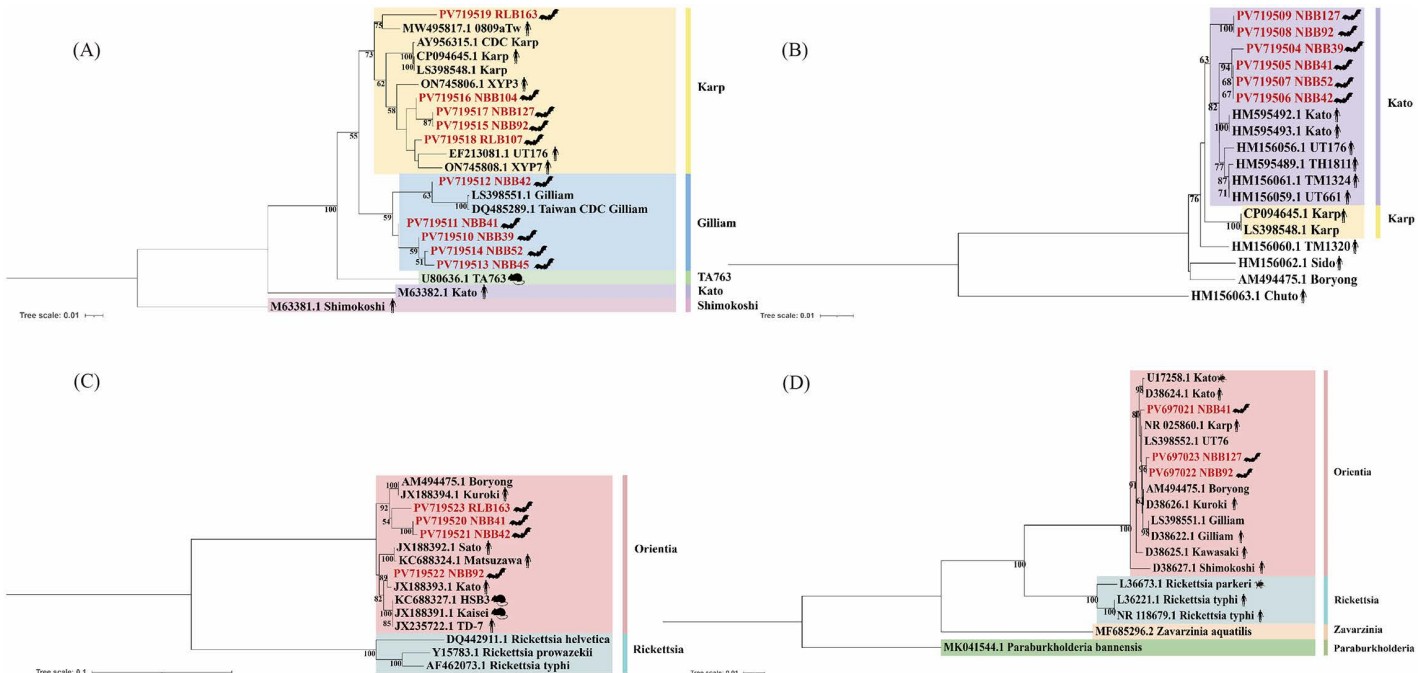

**Fig 1. Phylogenetic analysis of Ot strains detected in bats.** Neighbor-joining trees were constructed based on the nucleotide sequences of four genes: **(A)** 56-kDa TSA, **(B)** the 47-kDa *htrA*, **(C)** GroEL, and **(D)** 16S rRNA. Sequences obtained in this study are highlighted in red. Reference strains from GenBank include known human and animal isolates from various geographic regions. Scale bars indicate genetic distance based on the number of substitutions per site.

## Quantitative analysis of Ot loads in different bats tissues

The qPCR targeting the 47-kDa *htrA* gene was used to assess the bacterial loads of Ot in different tissues of 10 naturally infected bat samples. The median copy numbers (copies/µL) from highest to lowest were as follows: heart (1069.76 copies/µL; range: 108.73–2936.48), kidney (234.77 copies/µL; range: 34.69–2376.86), spleen (122.41 copies/µL; range: 48.16–863.80), lung (113.43 copies/µL; range: 43.72–4408.34), rectum (46.99 copies/µL; range: 31.86–533.50), liver (43.91 copies/µL; range: 24.70–161.39), and brain (28.60 copies/µL; range 23.09–43.55) (Fig 2). Our results suggest that Ot displays tissue tropism, with the highest bacterial load observed in cardiac and renal tissues.

## Evolutionary rate and divergence time estimation

Using a relaxed molecular clock model with an uncorrelated lognormal distribution, the mean evolutionary rate of the 56-kDa TSA gene was estimated to be $3.81 \times 10^{-5}$ substitutions/site/year (95% highest posterior density [HPD]: $2.01 \times 10^{-5}$ –$5.80 \times 10^{-5}$). Molecular dating analysis indicates that the most recent common ancestor (tMRCA) of Ot emerged around 4140 BC. At the strain level, the divergence time of RLB163, isolated from frugivorous bat (*R. amplexicaudatus*), was estimated to be approximately 126 AD. In contrast, the divergence of NBB92 from the insectivorous bat (*H. armiger*) was dated to approximately 1876 AD. Strains NBB39 and NBB41 from insectivorous bat (*H. larvatus*) were estimated to have diverged around 1526 AD. Our findings indicate a deep evolutionary history of Ot with lineage-specific divergence events in different bat (Fig 3).

## Discussion

The overall prevalence of Ot in bats (7.32%) observed in this study was lower than the previously reported prevalence in rodents in Yunnan Province [27]. Worldwide, previous studies have shown that Ot has been identified in *Muridae*, *Soricidae*, *Cricetidae*, *Canidae*, *Artiodactyla*, and birds [31–33]. Additionally, Ot has been identified in two bat species, specifically *Eptesicus serotinus* and *Rhinolophus ferrumequinum*, in South Korea [32], whereas no evidence of Ot was found in

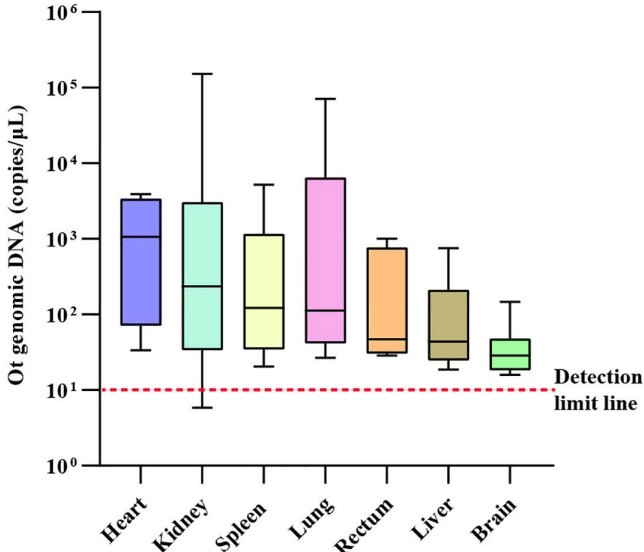

**Fig 2. Quantitative analysis of Ot DNA in different tissues of infected bat samples.** The bacterial loads in heart, kidney, spleen, lung, rectum, liver, and brain tissues of 10 Ot-positive bats were determined using quantitative PCR targeting the 47-kDa *htrA* gene. Results are expressed as median values with lower quartile and upper quartile (copies/µL). The red dashed line indicates the detection limit of the qPCR assay used in this study.

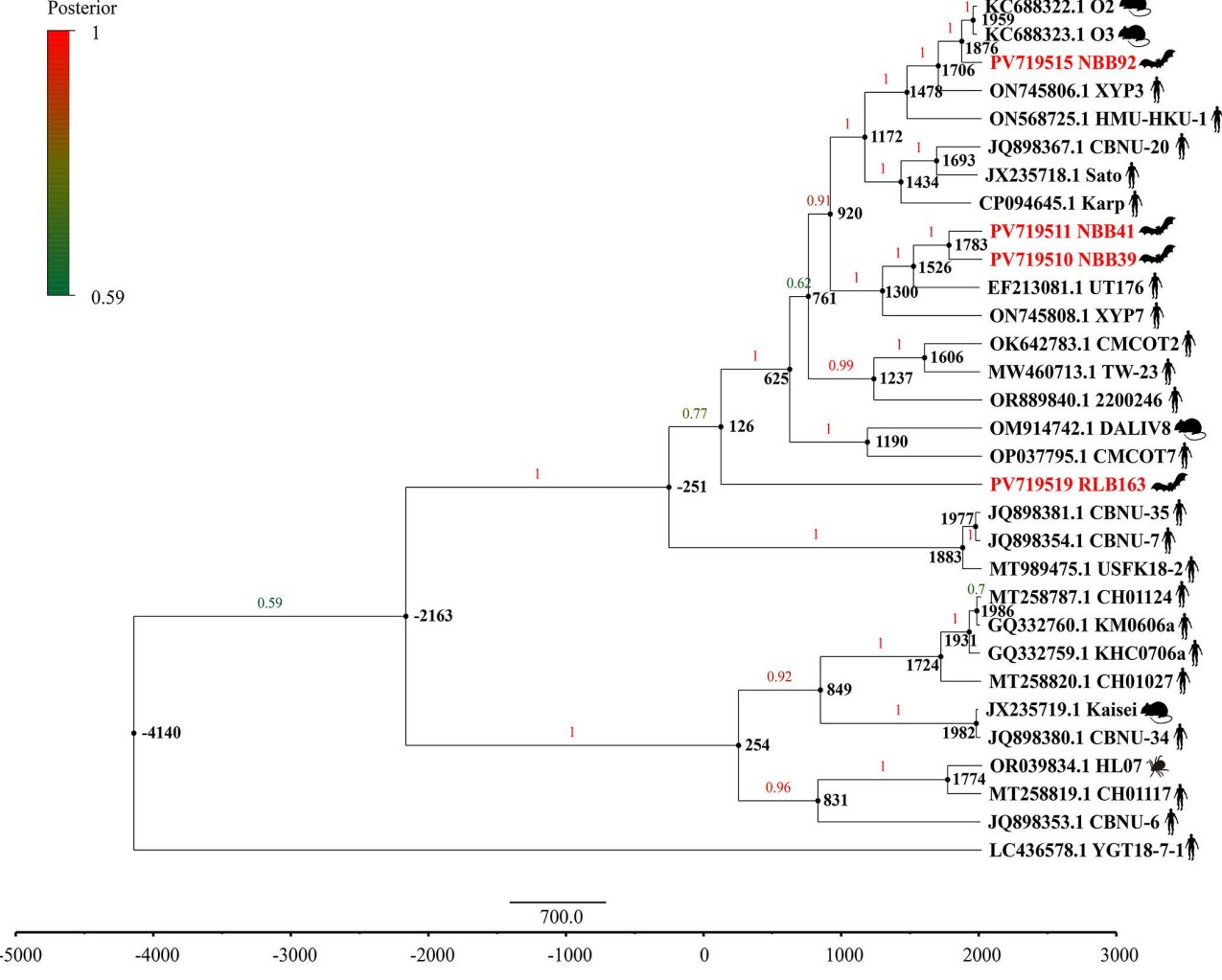

**Fig 3. Maximum clade credibility (MCC) tree with estimated divergence times based on the 56-kDa TSA gene sequences of Ot.** The tree was generated under a relaxed molecular clock model using BEAST v1.10.4. Red-labeled taxa represent sequences identified in this study. Node bars indicate the 95% highest posterior density (HPD) intervals for divergence time estimates.

*Eptesicus fuscus* in the United States [37]. In our study, Ot was identified in both insectivorous bats (*H. larvatus* and *H. armiger*) and frugivorous bats (*C. sphinx* and *R. amplexicaudatus*), indicating that Ot infected bats with diverse ecological niches.

Trombiculid mites are currently recognized as both vector and reservoir of Ot [41]. Vertical transmission through transovarial inheritance allows Ot to persist across mite generations, while chiggers can transmit Ot to rodents or humans through biting [42]. The Ot infections observed in bats in this study were likely due to bites by Ot-carrying chiggers. Notably, over 397 species of chigger mites have been reported to parasitize bats globally [40,43], and species such as *Leptotrombidium deliense*, *Leptotrombidium fletcheri*, and *Neotrombicula japonica* are known vectors of Ot [32,44]. As other small animals, bats in this study should be regarded as accidental dead-end hosts of Ot. Previous research has established that bats exhibit seasonal migratory behavior [45,46], including cross-sea and cross-habitat movements [47,48]. Based on these, we hypothesize that bats may carry infective chiggers, potentially transporting

Ot-positive chiggers to new areas and thereby serving as a vector-dispersing agent for Ot transmission. The migratory habits may play a role in connecting separate endemic foci. To determine whether they contribute to the long-range dispersal of Ot, our future work will investigate the species diversity, abundance, and prevalence of trombiculid mites parasitizing these bats.

Genotyping revealed the presence of at least three Ot genotypes, including Karp, Gilliam, and Kato, among strains isolated from bats along the China-Myanmar border. Phylogenetic analysis based on the 56-kDa TSA gene identified two genotypes: *H. larvatus* was infected with the Gilliam genotype, while *H.* armiger, *R. amplexicaudatus*, and *C. sphinx* carried the Karp genotype. In contrast, analysis based on the 47-kDa *htrA* gene showed that all detected strains belonged to the Kato genotype. This discrepancy may indicate that the clades of these *Orientia* are not sufficiently determined.

The qPCR analysis of Ot-infected bats revealed the presence of Ot DNA in multiple tissues. Ot was detected in brain tissue at a rather low load, consistent with observations in rodents [49]. Ot has tropism mainly for endothelial cells and less so for macrophages [50], and these cells infected with Ot are present in all organs. We therefore speculate that the bacterial loads are likely also affected by circulating Ot in the blood. In valid animal models, the lungs are a major target [51], but our study failed to yield the same result, possibly due to the small size of our positive samples. In the future, we will use more live-captured bats and perform blood sampling to better quantify Ot loads.

Molecular clock analysis of the 56-kDa TSA gene estimated the divergence of tMRCA of Ot to around 4,140 BC, predating the estimated divergence of strains isolated from bats. The earliest divergence among Ot strains isolated from bats was estimated around 126 AD. Our findings suggest that Ot strains detected in bats likely originated from ancestral lineages already circulating in rodents, humans, and mites, and the presence of Ot in bats has a long-standing evolutionary history. But it is worth noting that the molecular clock results presented should not be interpreted as reflecting the specific evolution of bat dead-end infections, but merely as a preliminary reference for understanding the evolutionary relationships between strains isolated from bats and strains from other hosts.

## Materials and methods

### Ethics statement

The relevant materials involving animal biomedical research were reviewed by the Medical Ethics Committee of Dali University, and it is considered that the project conforms to medical ethics (2021-PZ-177).

### Sample collection and processing

From July 2022 to August 2023, bat samples were collected from Ruili City, Yingjiang County, and Gengma County along the China-Myanmar border (Fig 4). In these regions, bats frequently forage on local economic crops such as fruits. To protect the crops, residents often install protective nets around plantations to deter bats activity. The bat specimens used in this study were collected opportunistically by retrieving naturally deceased individuals found on or around these protective nets. Initial species identification was performed based on morphological characteristics, followed by molecular confirmation through sequence analysis of the mitochondrial cytochrome b (mt-Cytb) gene [52–55]. After dissection, tissue samples including heart, liver, spleen, lung, kidney, rectum, and brain, were collected and placed into 2 mL cryogenic vials (CORNING, Shanghai, China). The samples were temporarily preserved in liquid nitrogen on-site and subsequently transported to the laboratory, where they were stored at −80°C until further analysis.

### DNA extraction

Under aseptic conditions, approximately 1 g of heart, liver, spleen, lung, kidney, rectum, and brain tissue samples were placed into GeneReady Animal PIII crushing tubes (Life Real, Hangzhou, China), followed by the addition of 600 μL of sterile phosphate-buffered saline (PBS). The samples were homogenized using a GeneReady Ultimate grinder (Life Real, Guangzhou, China). Subsequently, 300 μL of the resulting supernatant was used for DNA extraction with a commercial

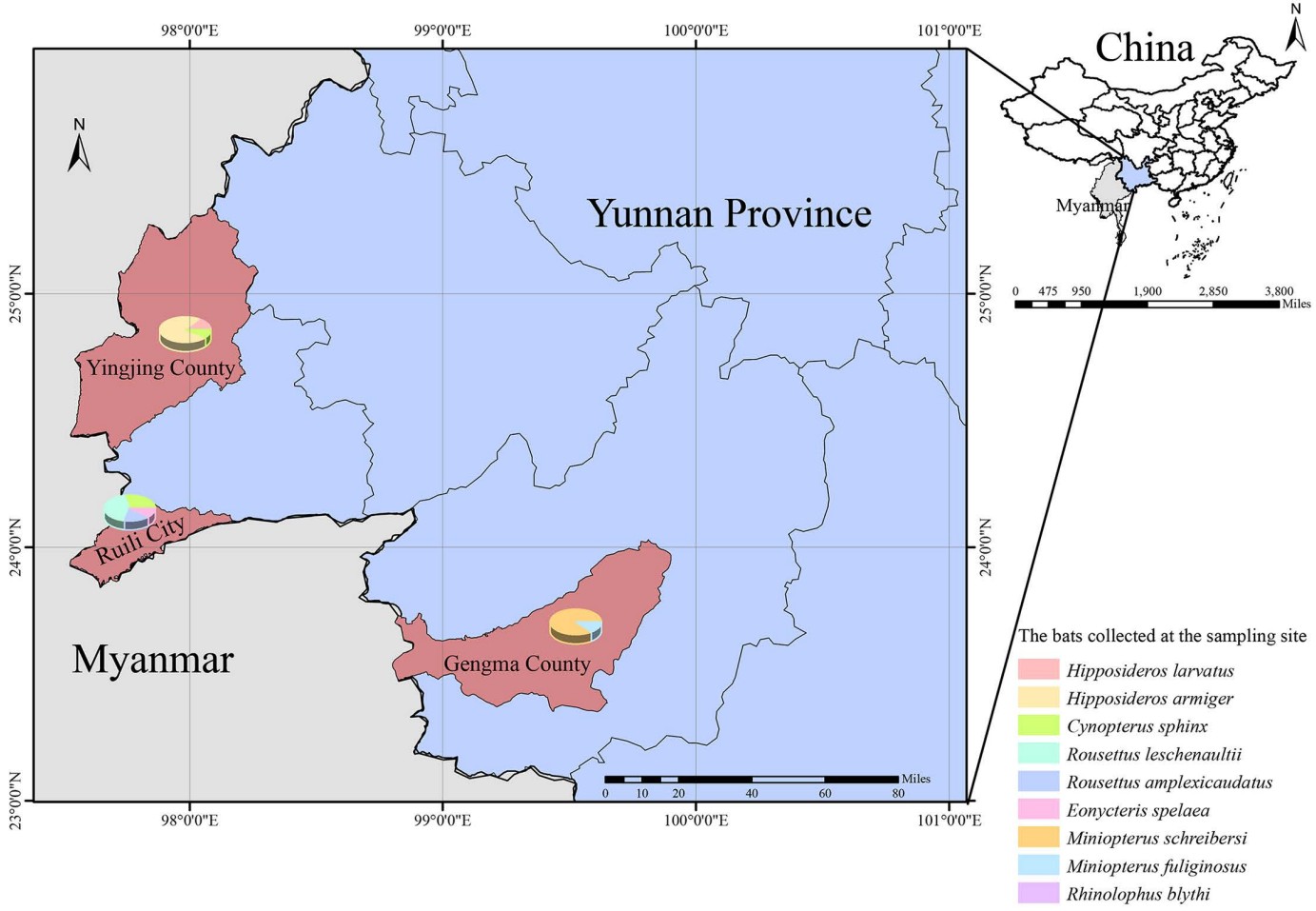

**Fig 4. Sampling locations.** The map on the upper right shows the geographic locations of China and Myanmar. The enlarged map on the left highlights the three sampling sites in Yunnan Province: Ruili City, Yingjiang County, and Gengma County, located along the China-Myanmar border. The maps were generated using QGIS Desktop 3.0.1 software (vector map: geoBoundaries: https://www.geoboundaries.org/globalDownloads.html). The license: https://creativecommons.org/licenses/by/4.0/.

DNA extraction kit (TIANGEN, Beijing, China) using a fully automated nucleic acid extraction and purification system (BIOER, Hangzhou, China). Extracted DNA samples were stored at −80°C for downstream analysis.

### Detection of Ot

A real-time quantitative PCR (qPCR) assay targeting the 47-kDa *htrA* gene of Ot was used, as previously described by our research team [27]. The 20 µL reaction mixture included 10 µL of 2 × Probe Master Mix V2, 0.4 µL each of forward and reverse primers, 0.8 µL of probe, 7.4 µL of RNase-Free ddH$_2$O, and 1 µL of DNA template (primer and probe sequences provided in S2 Table). The qPCR was performed using the Applied Biosystems 7500 Real-Time PCR System (Thermo Fisher Scientific, Waltham, MA, USA).

Samples testing positive by qPCR were further analyzed using nested PCR (nPCR) to amplify four genes: 56-kDa TSA, 47-kDa *htrA*, GroEL, and 16S rRNA (S2 Table). Each round of PCR was conducted in a 25 µL reaction system containing 12.5 µL of 2 × Rapid Taq Master Mix, 1 µL each of forward and reverse primers, 9.5 µL of RNase-Free H$_2$O, and 1 µL of

DNA template. Amplification products were assessed by agarose gel electrophoresis. Brands of the expected size were excised and purified using a gel extraction kit (OMEGA Bio-tek, Norcross, GA, USA), and sent to Sangon Biotech Co., Ltd. (Shanghai, China) for bidirectional Sanger sequencing.

### Full-length amplification of 56-kDa TSA, 47-kDa *htrA*, GroEL, and 16S rRNA genes

Using DNA extracted from spleen tissues of nPCR-positive bats, the full-length sequences of four target genes (56-kDa TSA, 47-kDa *htrA*, GroEL, and 16S rRNA) were amplified and sequenced. Gene-specific primers were designed based on multiple sequence alignments of Ot sequences from GenBank using Clustal X2 software, with additional primers designed iteratively based on initial and subsequent sequencing results (S3 Table). PCR products of the expected sizes were purified and sequenced. Resulting reads were assembled using Geneious Prime software v2025.0.3 to generate the full-length sequences.

### Sequence identification and phylogenetic analysis

Obtained sequences were compared with publicly available sequences using the BLAST tool (https://blast.ncbi.nlm.nih.gov/Blast.cgi, accessed on September 25, 2024). Reference sequences of 56kDa TSA, 47kDa *htrA*, GroEL, and 16S rRNA genes were downloaded from GenBank. Sequence identity analysis was performed using BioAider software v1.423 [56]. Multiple sequence alignment was conducted using Clustal X2 software, and phylogenetic trees were constructed using the Neighbor-Joining method with 1,000 bootstrap replicates [57]. Phylogenetic trees were visualized using iTOL software (https://itol.embl.de/, accessed on May 11, 2025). All sequences generated in this study have been submitted to GenBank (accession numbers: PV719504–PV719533, and PV697021–PV697023).

### Quantitative analysis of Ot in different bat tissues

Quantitative detection of Ot in various tissues (heart, liver, spleen, lung, kidney, rectum, brain) from nPCR-positive bats was performed by qPCR. Results are presented as median values with interquartile ranges (lower quartile, upper quartile).

### Estimation of divergence dates

Following a preliminary GenBank search for "*Orientia tsutsugamushi*", we selected sequences containing the full-length 56-kDa TSA gene, along with host and collection date metadata. For the molecular clock analysis, we included only a single sequence from groups of highly similar sequences sharing identical sampling location, date, and host.

Multiple sequence alignment of the full-length 56-kDa TSA gene sequences was conducted using Clustal X2 software. The temporal signal in the aligned sequences was assessed with TreeTime program [58]. A weak correlation between sampling time and genetic distance was observed (correlation coefficient [$R^2$] = $1.8727 \times 10^{-2}$). A prior evolutionary rate of $3 \times 10^{-5}$ to $6 \times 10^{-5}$ substitutions/site/year was applied. Bayesian phylogenetic analysis was performed using BEAST v1.10.4 [59], with a Markov chain run for 10 million steps, sampling every 1,000 steps. An uncorrelated lognormal relaxed molecular clock model was employed to estimate the mean evolutionary rate and the time to the most recent common ancestor (tMRCA). Convergence and effective sample size (ESS > 200) were verified using Tracer program v1.6. A maximum clade credibility (MCC) tree was constructed after discarding by the first 10% of states using TreeAnnotator package [60].

### Statistical analysis

Statistical analysis was performed using SPSS software v25.0 (IBM Corp., Armonk, NY, USA). The chi-square ($\chi^2$) test was used to compare detection rates between nPCR and qPCR. A *P*-value < 0.05 was considered statistically

significant. The map was generated in QGIS Desktop 3.0.1 software using vector map data obtained from the geo-Boundaries datasets [61].

## Supporting information

**S1 Table. Sequences obtained in this study.**
(XLSX)

**S2 Table. The primers for the detection of Ot in this study.**
(DOCX)

**S3 Table. Primers used to amplify the 56-kDa TSA, 47-kDa *htrA*, GroEL, and 16S rRNA genes of Ot in bats.**
(XLSX)

**S1 Fig. Identity comparisons of nucleotide and amino acid sequences of Ot 56-kDa TSA gene in bats from this study.** Top right indicates acid sequence identity, while bottom left shows nucleotide sequence identity.
(TIF)

**S2 Fig. Identity comparisons of nucleotide and amino acid sequences of Ot 47-kDa *htrA* gene in bats from this study.** Top right indicates acid sequence identity, while bottom left shows nucleotide sequence identity.
(TIF)

**S3 Fig. Identity comparisons of nucleotide and amino acid sequences of Ot GroEL gene in bats from this study.** Top right indicates acid sequence identity, while bottom left shows nucleotide sequence identity.
(TIF)

**S4 Fig. Identity comparisons of nucleotide sequences of Ot 16S rRNA gene in bats from this study.**
(TIF)

## Acknowledgments

We are deeply grateful for the assistance provided by the Centre for Disease Control and Prevention (CDC) of Yingjiang County, CDC of Ruili City, and CDC of Gengma County during the sampling process.

## Author contributions

**Conceptualization:** Yunzhi Zhang.

**Data curation:** Yun Long, Jiawei Tian, Peiyu Han, Song Wu, Yuhong Chen, Wanchun Cao, Lijun Guo.

**Formal analysis:** Yun Long, Jiawei Tian, Peiyu Han, Song Wu, Lidong Zong, Chenjie He, Yuhong Chen.

**Funding acquisition:** Yunzhi Zhang.

**Investigation:** Yun Long, Jiawei Tian, Peiyu Han, Song Wu, Lidong Zong, Chenjie He, Yuhong Chen, Wanchun Cao, Yunzhi Zhang.

**Methodology:** Yun Long, Jiawei Tian, Peiyu Han, Yunzhi Zhang.

**Project administration:** Bo Wang, Lijun Guo, Yunzhi Zhang.

**Resources:** Yun Long, Jiawei Tian, Peiyu Han, Song Wu, Lidong Zong, Chenjie He, Yuhong Chen.

**Software:** Yun Long, Jiawei Tian, Peiyu Han, Wanchun Cao.

**Supervision:** Bo Wang, Lijun Guo, Yunzhi Zhang.

**Validation:** Yun Long, Jiawei Tian, Peiyu Han.

**Visualization:** Yun Long, Jiawei Tian, Peiyu Han.

**Writing – original draft:** Yun Long, Jiawei Tian, Yunzhi Zhang.

**Writing – review & editing:** Bo Wang, Lijun Guo, Yunzhi Zhang.

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
