## [Decision Letter · Decision Letter 0]

29 Sep 2025

First discovery of bat-borne Orientia tsutsugamushi and its origin

Dear Dr. Zhang,

Thank you for submitting your manuscript to PLOS Neglected Tropical Diseases. After careful consideration, we feel that it has merit but does not fully meet PLOS Neglected Tropical Diseases's publication criteria as it currently stands. Therefore, we invite you to submit a revised version of the manuscript that addresses the points raised during the review process.

Please submit your revised manuscript within 60 days Nov 28 2025 11:59PM. If you will need more time than this to complete your revisions, please reply to this message or contact the journal office at plosntds@plos.org. Please include the following items when submitting your revised manuscript:

We look forward to receiving your revised manuscript.

Kind regards,

Nam-Hyuk Cho

Academic Editor

Elsio Wunder Jr

Section Editor

Shaden Kamhawi

co-Editor-in-Chief

Paul Brindley

co-Editor-in-Chief

**Additional Editor Comments (if provided):**

**Journal Requirements:**

**Reviewers' Comments:**

Reviewer's Responses to Questions

**Key Review Criteria Required for Acceptance?**

**Methods**

-Are the objectives of the study clearly articulated with a clear testable hypothesis stated?

-Is the study design appropriate to address the stated objectives?

-Is the population clearly described and appropriate for the hypothesis being tested?

-Is the sample size sufficient to ensure adequate power to address the hypothesis being tested?

-Were correct statistical analysis used to support conclusions?

-Are there concerns about ethical or regulatory requirements being met?

Reviewer #1: (No Response)

Reviewer #2: There are inconsistencies between the Methods and Results sections; for instance, the reported target sequence length does not match the obtained sequence length that was subsequently analyzed. This discrepancy should be clarified. In addition, the description of the nested PCR procedure is insufficient. Providing detailed methodological information is essential to ensure reproducibility of the study.

Reviewer #3: See attachment

**Results**

-Does the analysis presented match the analysis plan?

-Are the results clearly and completely presented?

-Are the figures (Tables, Images) of sufficient quality for clarity?

Reviewer #1: The authors present their findings as if bats play a role in the zoonotic cycle of Orientia tsutsugamushi. Bats and all other animal hosts are accidental dead-end hosts of Orientia tsutsugamushi. Orientia are maintained solely by transovarian transmission in mites. The concept that strains of Orientia have evolved in bats is inaccurate.

The author's concept that Orientia is primarily transmitted by chiggers suggests that the other modes of transmission, which include only transplacentalal, transfusion, and organ transplantation, have greater importance than they do.

The fact that analysis of different genes puts the same strain in different clades indicates that the clades of these Orientia are not sufficiently determined and the analyses are not useful.

The low number of copies of the 47 kDa gene in the organs suggests that some organs were probably only measuring Orientia in the blood in the organ rather than infection of the cells in the organ. The authors do not report the concentration of Orientia in the blood.

Lines 208-210: Small mammals do not play a role in the transmission of scrub typhus. They are merely a source of blood for the larval mites.

The molecular clock analysis is unlikely to reflect evolution related to the accidental dead-end infection of bats.

Reviewer #2: There are inconsistencies between the Methods and Results sections; for instance, the reported target sequence length does not match the obtained sequence length that was subsequently analyzed. This discrepancy should be clarified. In addition, the description of the nested PCR procedure is insufficient. Providing detailed methodological information is essential to ensure reproducibility of the study.

Reviewer #3: See attachment

**Conclusions**

-Are the conclusions supported by the data presented?

-Are the limitations of analysis clearly described?

-Do the authors discuss how these data can be helpful to advance our understanding of the topic under study?

-Is public health relevance addressed?

Reviewer #1: Please see comments in the results section

Reviewer #2: The manuscript does not describe the habitat range of the studied bat species or their potential ecological interactions with chigger mites. Such information is crucial to evaluate the role of bats in long-distance dispersal of O. tsutsugamushi and would significantly strengthen the novelty and conclusions of the study.

Furthermore, the conclusions drawn from the molecular clock analysis may be premature, as they are based on a limited number of sequences, including some that are not clearly described. A more detailed explanation of the sequence selection and validation would help to strengthen this part of the study.

Reviewer #3: See attachment

**Editorial and Data Presentation Modifications?**

Reviewer #1: (No Response)

Reviewer #2: (No Response)

Reviewer #3: See attachment

**Summary and General Comments**

Reviewer #1: If indeed this is the first detection of Orientia in bats, it is an interesting observation.

Reviewer #2: (No Response)

Reviewer #3: See attachment

PLOS authors have the option to publish the peer review history of their article (what does this mean? ). If published, this will include your full peer review and any attached files.

**Do you want your identity to be public for this peer review?** For information about this choice, including consent withdrawal, please see our Privacy Policy .

Reviewer #1: No

Reviewer #2: No

Reviewer #3: No

**Figure resubmission:**
---

## [Decision Letter · Decision Letter 1]

2 Dec 2025

Response to Reviewers
Revised Manuscript with Track Changes
Manuscript

Shaden Kamhawi

co-Editor-in-Chief

Paul Brindley

co-Editor-in-Chief

**Additional Editor Comments (if provided):**
**Journal Requirements:**

**Reviewers' comments:**

**Key Review Criteria Required for Acceptance?**

**Methods**

-Are the objectives of the study clearly articulated with a clear testable hypothesis stated?

-Is the study design appropriate to address the stated objectives?

-Is the population clearly described and appropriate for the hypothesis being tested?

-Is the sample size sufficient to ensure adequate power to address the hypothesis being tested?

-Were correct statistical analysis used to support conclusions?

-Are there concerns about ethical or regulatory requirements being met?

Reviewer #1: The methods are satisfactory.

Reviewer #3: (No Response)

**Results**

-Does the analysis presented match the analysis plan?

-Are the results clearly and completely presented?

-Are the figures (Tables, Images) of sufficient quality for clarity?

Reviewer #1: The results are valid.

Reviewer #3: (No Response)

**Conclusions**

-Are the conclusions supported by the data presented?

-Are the limitations of analysis clearly described?

-Do the authors discuss how these data can be helpful to advance our understanding of the topic under study?

-Is public health relevance addressed?

Reviewer #1: The authors have responded appropriately in part to the previous critiques. However, the text is inconsistent in its responses and needs to be harmonized.

1. Lines 19 and 40: It is misleading to state that Orientia tsutsugamushi is mainly or primarily transmitted by feeding chiggers. It is nearly exclusively chigger-transmitted. Deletion of the words "mainly" and "primarily" would be appropriate.

2. Lines 32, 47: The term "bat-derived" could be interpreted as Orientia tsutsugamushi having evolved in bats. This is not true. The strains are mite-derived. It would be more appropriate to state "Orientia tsutsugamushi isolated from bats" than "bat-derived".

3. Lines 34, 46, 208-214: Orientia tsutsugamushi has tropism mainly for endothelial cells and less so for macrophages. These cells infected with Orientia tsutsugamushi are present in all organs. The bacterial loads described are likely also affected by circulating Orientia tsutsugamushi in blood. In valid animal models the lungs are a major target

4. Lines 35-37: The findings reported in this manuscript do not raise new questions regarding the role of bats in transmission. Bats play no role in transmission of Orientia tsutsugamushi.

5. Lines 89-90: It should be noted that Orientia chuto has been detected in southwestern Asia and eastern Africa.

6. Lines 99-101: Rodents and other mammals are not reservoirs of Orientia tsutsugamushi. They are dead-end hosts.

7. Lines 186 and 188: It would be appropriate to remove the designation China. Taiwan is sufficient.

8. Line 237: Chiggers feed on tissue fluid from the dermis, not blood.

Reviewer #3: Discussion

L219: “demonstrate the detection” = bad style. The sentence refers to China or worldwide? Not clear how this sentence connects to the previous…

L221: “Specifically” is unclear wording here. Explain that you are talking about bats…

L225: “can infect”, replace by “infected”

L226: Chiggers do not feed on blood. The entire sentence seems not to make much sense. Better erase the sentence starting with “While small…”.

L231: how to conclude that the bites were accidental? You do not know the mite species involved.

L235: Start sentence with “As other small animals, bats…“

L238: erase “for dispersal”

L253ff: erase the sentence “the associated…”. There are no CNS manifestations in bats and the findings in dead bats cannot be used to explain human disease in a half sentence.

**Editorial and Data Presentation Modifications?**

Reviewer #1: See the above comments on conclusions

Reviewer #3: (No Response)

**Summary and General Comments**

Reviewer #1: See the above comments on conclusions

Reviewer #3: The article describes the detection of Orientia tsutsugamushi (Ot) in bat tissue samples. Since information on the infection of non-rodent animals are scarce, the presented data is scientifically relevant.

The following comments refer to the R1 version of the manuscript. Line numbers refer to the word doc “PNTD-D-25-01569_Manuscript_Clean with Highlights”.

Author summery

L39: Please erase “But”, since there no contrast to the sentence before.

Introduction

L54: “having also been documented” can be erased to improve style

L81: the word “expanding” is somehow misleading. Better say “…appears to be wider than previously known, as Orientia species have been detected in the Middle East, Africa, and Chile (20-22)”. Erase “suggesting a broader…”.

L91ff: unclear wording mixing hosts for Ot with host for chiggers. I suggest: L91 replace “natural” by “potential”. And “This” by “The”.

L93ff: you have to clarify “critical role”. For what? Small animals are not the primary reservoir for Ot but maintain chigger populations (the primary Ot reservoir). So they get exposed (serology) and infected (PCR) with Ot. Sorry, this part has to be re-written. The reptiles sentence can be erased.

PLOS authors have the option to publish the peer review history of their article (what does this mean? ). If published, this will include your full peer review and any attached files.

**Do you want your identity to be public for this peer review?** For information about this choice, including consent withdrawal, please see our Privacy Policy .

Reviewer #1: No

Reviewer #3: No

**Figure resubmission:**

**Reproducibility:** To enhance the reproducibility of your results, we recommend that authors of applicable studies deposit laboratory protocols in protocols.io, where a protocol can be assigned its own identifier (DOI) such that it can be cited independently in the future. Additionally, PLOS ONE offers an option to publish peer-reviewed clinical study protocols. Read more information on sharing protocols at https://plos.org/protocols?utm_medium=editorial-email&utm_source=authorletters&utm_campaign=protocols

---

## [Editor Report · Decision Letter 2]

15 Dec 2025

Dear Dr. Zhang,

We are pleased to inform you that your manuscript 'Molecular detection of Orientia tsutsugamushi infection in bats from the China-Myanmar border' has been provisionally accepted for publication in PLOS Neglected Tropical Diseases.

Best regards,

Nam-Hyuk Cho

Academic Editor

Elsio Wunder Jr

Section Editor

Shaden Kamhawi

co-Editor-in-Chief

Paul Brindley

co-Editor-in-Chief

---

## [Editor Report · Acceptance letter]

Dear Dr. Zhang,

We are delighted to inform you that your manuscript, "Molecular detection of Orientia tsutsugamushi infection in bats from the China-Myanmar border," has been formally accepted for publication in PLOS Neglected Tropical Diseases.

Best regards,

Shaden Kamhawi

co-Editor-in-Chief

Paul Brindley

co-Editor-in-Chief
